# Unravelling subclonal heterogeneity and aggressive disease states in TNBC through single-cell RNA-seq

Mihriban Karaayvaz[1], Simona Cristea[2,3,4], Shawn M. Gillespie[1,5], Anoop P. Patel[6], Ravindra Mylvaganam[1,5], Christina C. Luo [1,5], Michelle C. Specht[7], Bradley E. Bernstein[1,5,8,9], Franziska Michor [2,3,4,8,9,10] & Leif W. Ellisen[1]

Triple-negative breast cancer (TNBC) is an aggressive subtype characterized by extensive intratumoral heterogeneity. To investigate the underlying biology, we conducted single-cell RNA-sequencing (scRNA-seq) of >1500 cells from six primary TNBC. Here, we show that intercellular heterogeneity of gene expression programs within each tumor is variable and largely correlates with clonality of inferred genomic copy number changes, suggesting that genotype drives the gene expression phenotype of individual subpopulations. Clustering of gene expression profiles identified distinct subgroups of malignant cells shared by multiple tumors, including a single subpopulation associated with multiple signatures of treatment resistance and metastasis, and characterized functionally by activation of glycosphingolipid metabolism and associated innate immunity pathways. A novel signature defining this subpopulation predicts long-term outcomes for TNBC patients in a large cohort. Collectively, this analysis reveals the functional heterogeneity and its association with genomic evolution in TNBC, and uncovers unanticipated biological principles dictating poor outcomes in this disease.

[1] Center for Cancer Research, Massachusetts General Hospital and Harvard Medical School, Boston, MA 02114, USA. [2] Department of Biostatistics and Computational Biology, Dana-Farber Cancer Institute, Boston, MA 02115, USA. [3] Department of Biostatistics, Harvard T. H. Chan School of Public Health, Boston, MA 02215, USA. [4] Department of Stem Cell and Regenerative Biology, Harvard University, Cambridge, MA 02138, USA. [5] Department of Pathology, Massachusetts General Hospital and Harvard Medical School, Boston, MA 02114, USA. [6] Department of Neurosurgery, Massachusetts General Hospital and Harvard Medical School, Boston, MA 02114, USA. [7] Department of Surgical Oncology, Massachusetts General Hospital and Harvard Medical School, Boston, MA 02114, USA. [8] The Broad Institute of Harvard and MIT, Cambridge, MA 02139, USA. [9] The Ludwig Center at Harvard, Boston, MA 02215, USA. [10] Center for Cancer Evolution, Dana-Farber Cancer Institute, Boston, MA 02115, USA. These authors contributed equally: Mihriban Karaayvaz, Simona Cristea. Correspondence and requests for materials should be addressed to F.M. (email: michor@jimmy.harvard.edu) or to L.W.E. (email: lellisen@mgh.harvard.edu)

Triple-negative breast cancer, defined clinically as lacking estrogen receptor (ER) and progesterone receptor (PgR) expression as well as human epidermal growth factor receptor 2 (HER2) gene amplification, represents up to 20% of all breast cancers and is associated with a more aggressive clinical course compared to other breast cancer subtypes[1,2]. The majority of TNBCs share common histological and molecular features including frequent p53 mutation, a high proliferative index, and frequent expression of a basal-like gene expression signature[3]. Nonetheless, TNBC is a disease entity characterized by extensive inter-tumor as well as intra-tumor heterogeneity, and likely represents multiple clinically and biologically distinct subgroups that have not yet been clearly defined[4,5].

Deep sequencing of tumor-associated somatic mutations has revealed a substantial level of intratumoral heterogeneity of TNBC[3], while multi-region sequencing showed that a particularly large extent of spatial subclonal diversification is associated with TNBC compared to other breast cancer subtypes[6]. Single-nucleus genome sequencing yielded similar observations and together with mathematical modeling suggested a mutation rate within ER + tumors close to that of normal cells, while TNBC exhibited a rate approximately 13-fold higher[7]. Thus, TNBC is uniquely characterized by persistent intratumoral diversification.

Multiple lines of evidence suggest that the intratumoral diversity of TNBC is not only a driver of pathogenesis, but also of treatment resistance, metastasis, and poor clinical outcomes[8]. While most primary TNBCs exhibit substantial responses to pre-operative chemotherapy, a failure to achieve complete elimination of viable tumor cells in the breast (so-called pathologic complete response) is associated with very poor outcomes in TNBC but not in ER+ breast cancers[9,10]. Therefore, unlike in ER+ cancers, killing the majority of the bulk population of TNBC cells has relatively little impact on outcomes. This finding implies that a minor subpopulation of TNBC cells is responsible for metastatic dissemination. Clonal evolution within the primary tumor is a likely driver of this process, as multi-site metastases in TNBC can be attributed to multiclonal seeding from individual clones that are identifiable in the primary tumor[11]. Given that most studies of human tumors are limited to bulk analysis, however, the existence and precise nature of subclonal diversification, signaling, and cooperation in human breast cancer remains to be established.

A small number of studies have characterized the genomic diversity of TNBC at the single-cell level, revealing a pattern that reflects punctuated evolution of copy number variations during TNBC progression, followed by expansion of a dominant subclone[7,12]. While these findings imply that such subclones harbor properties driving their selective advantage, DNA-based analyses alone have been unable to elucidate the cell states and fates that underlie this process. To address this issue, we conducted single-cell RNA-sequencing on >1500 cells from six freshly collected, untreated primary TNBC tumors. Through detailed computational analyses of individual tumor cells and the subpopulations they encompass, we reveal the phenotypes and biology underlying the genetic evolution and clinical behavior of TNBC.

## Results

### Acquisition of scRNA-seq profiles from primary TNBC.
In order to understand intercellular heterogeneity in TNBC, we collected tumors from six women presenting with primary, non-metastatic triple-negative invasive ductal carcinomas prior to any local or systemic therapy. Assessment of ER/PR/HER2-negative status was performed using strict clinical and histological criteria (Supplementary Table 1). All tumors were histologically characteristic of TNBC, comprised of a dense mass of invasive ductal carcinoma cells with variable infiltration of immune and stromal elements (Supplementary Fig. 1). Of six tumors with sufficient tissue for analysis, two (tumors 84 and 126) were associated with local axillary lymph node involvement (Supplementary Table 1).

Fresh tumors underwent rapid dissociation followed by flow-cytometry sorting of viable single cells. To capture the full spectrum of tumor cellular composition, we sorted a subset of cells and tumors with no pre-selection, and to ensure adequate numbers of malignant cells for analysis, we sorted another subset following depletion of immune cells based on CD45 staining (Fig. 1a). Individual cells underwent preparation of cDNA and library construction, followed by next-generation sequencing (NGS). After stringent quality control and normalization, we analyzed a total of 1189 cells ranging among patients from 78 cells (tumor 58) to 286 cells (tumor 89) (Fig. 1a).

### Cellular heterogeneity within primary TNBCs.
Our first analysis involved identifying distinct cell populations within the tumors using a multi-step approach involving marker genes together with clustering (Fig. 1b, c; Supplementary Methods). This combination approach was designed to provide a more robust identification of cell types than that achieved by using either methodology alone. Thus, we first evaluated the expression of specific sets of genes previously established to define immune, endothelial, and stromal cells, as well as key mammary epithelial subpopulations including basal cells, luminal progenitors, and mature luminal cells[13,14]. We then used clustering to refine the identified cell types. This combined analysis reliably classified 1112 of 1189 cells into non-epithelial ($n = 244$) and epithelial ($n = 868$) types. A subset of TNBCs are known to exhibit substantial immune cell infiltration, and as anticipated, CD45-unselected tumors contained large proportions of immune cells, the majority of which were T lymphocytes[15,16]. The next most prevalent immune cell subset were macrophages, which were present in each CD45-unselected tumor. Like immune cells, stromal elements are known to vary between TNBC cases[17], and we identified such cells as <15% of total cells in each tumor. Endothelial cells were a minority population, representing at most 4% of cells in one tumor (Fig. 1c). The most prominent epithelial cell population in each tumor expressed luminal cell and luminal progenitor markers, while in some tumors (e.g., tumor 89), a minority population was evident that expressed markers of myoepithelial cells including *ACTA2* and *TAGLN* (Fig. 1b).

As a first step to identifying malignant cells, we determined their cell cycle status using validated gene signatures previously shown to identify G1/S and G2/M cell cycle phases and distinguishing high cycling from low cycling cells[13]. This analysis revealed substantial variation among tumors in the proportion of cycling cells, ranging from <5% (tumor 58) to >34% (tumor 81, Fig. 1d). Notably, most cycling cells (98.5% of all cycling cells) were identified as epithelial, consistent with our cell type classification and suggesting that the malignant cells reside in this compartment (Fig. 1e; Supplementary Fig. 2). We validated these findings by immunohistochemical staining of tumor sections with Ki67, a widely used clinical marker of cycling cells[18]. Indeed, we observed a high correlation between the predicted proportion of cycling cells and the percentage of Ki67-positive cells within each tumor (Fig. 1e; Supplementary Fig. 2).

### Subclonal heterogeneity defines malignant TNBC cells.
To further support the classification of malignant versus non-malignant cells, we used complementary approaches based on (i) gene expression clustering, (ii) estimated copy number

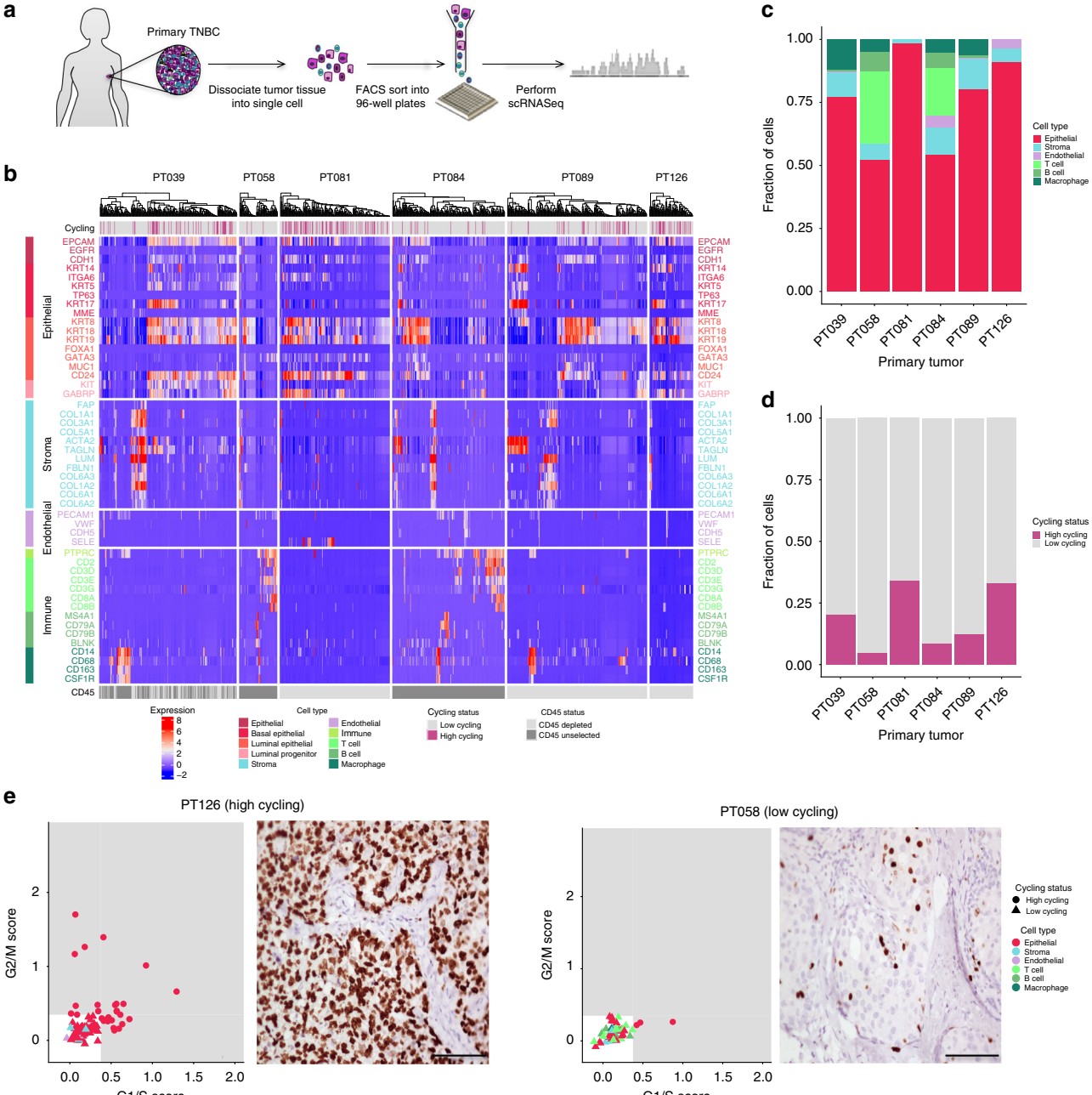

**Fig. 1** Intercellular heterogeneity in TNBC quantified by scRNA-seq. **a** Workflow showing collection and processing of fresh TNBC primary tumors for generating scRNA-seq data. **b** Heatmap of the 1189 cells that passed quality control, with columns representing cells and rows representing established gene expression markers for the cell types indicated on the left, clustered separately for each of the six TNBC cases. The upper bar denotes inferred high cycling (pink) and low cycling (gray) cells, as identified by quantifying the expression of a set of relevant genes (see Supplementary Methods). Bottom bar denotes cells collected in the presence/absence of CD45 + cell depletion. **c** Bar plot depicting the distribution of the 1112 cells assigned to specific cell types, by patient. **d** Bar plot depicting the high cycling/low cycling distribution of the 1112 cells, by patient. **e** Proliferation characteristics for two representative TNBC patients, depicted as either the inferred cycling status of single cells (left) or immunohistochemistry staining for Ki67 (right). Scale bars represent 50 μm. A cell is considered high cycling if it has high G1/S or G2/M scores, as identified by quantifying the expression of a set of relevant genes. The two ways of quantifying proliferation show good concordance

variations (CNVs), and (iii) intercellular heterogeneity of transcriptomes[13,19]. Analysis of all cells across patients using tSNE clustering[20] revealed substantial separation between non-epithelial populations, which were well-segregated into distinct clusters based on their cell type, and epithelial populations, which formed multiple subgroups (Fig. 2a). This pattern was most apparent when non-epithelial and epithelial cells were analyzed separately (Fig. 2b, c). In contrast to non-epithelial cells, epithelial cells generally separated into tumor-specific clusters (particularly tumors 39 and 81), but interestingly also into clusters with contribution of cells from multiple tumors (Fig. 2a–c). Prior single-cell analyses of melanoma and glioblastoma showed clustering of malignant cells primarily by patient[13,21], supporting a significant degree of inter-tumor heterogeneity. In contrast, in keeping with our findings, a recent single-cell analysis of breast cancers showed both patient-specific and shared clustering of malignant cells,

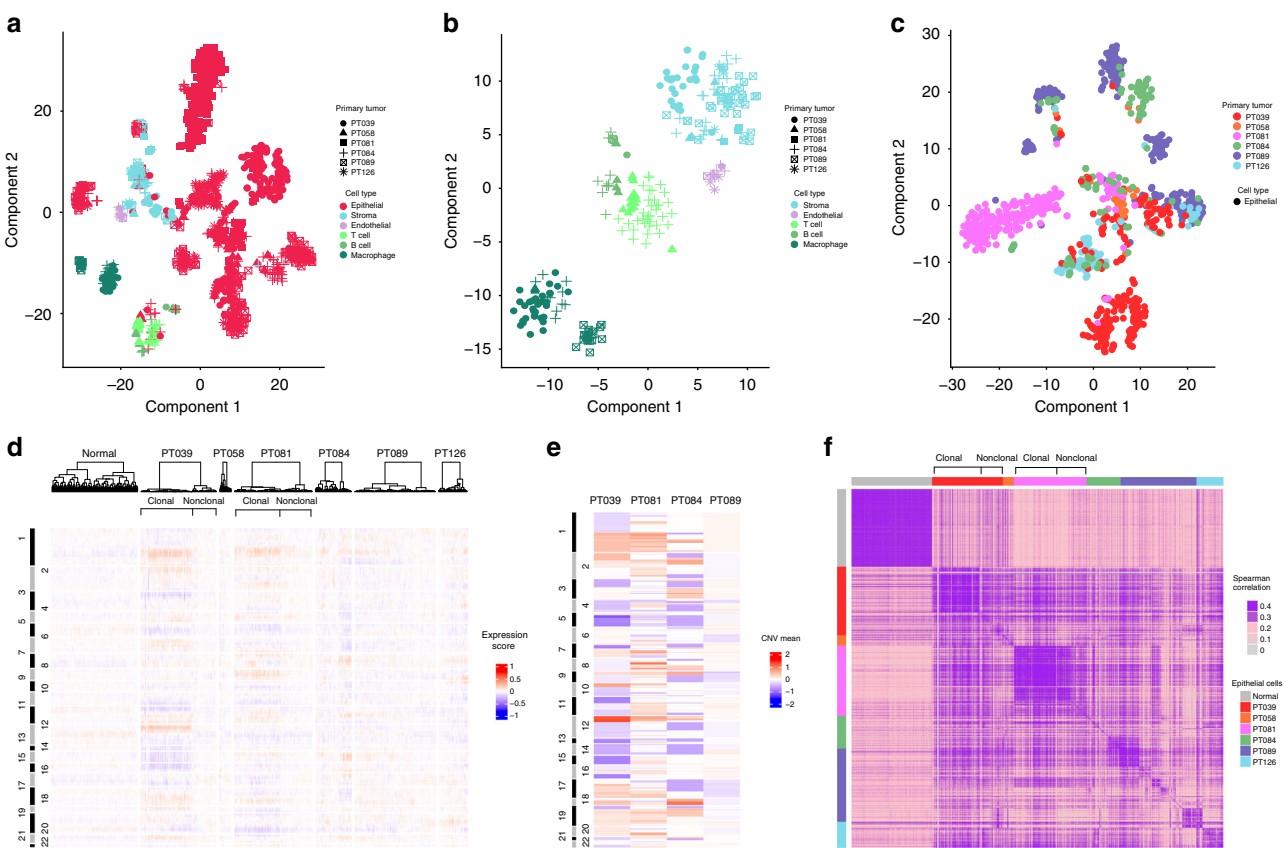

**Fig. 2** Clustering, genomic CNVs, and correlation maps classify most epithelial cells as malignant. **a** t-SNE plot of all 1112 classified cells, demonstrating separation of non-epithelial cells by cell type. **b** t-SNE plot of the 244 non-epithelial cells, demonstrating separation by cell type, and no distinguishable patient effect. **c** t-SNE plot of the 868 epithelial cells, showing mixed separation by patient, and substantial clustering of cells from different patients, suggesting pronounced intra-tumor heterogeneity. **d** Inferred CNVs from the single-cell gene expression data. Columns represent individual cells, and rows represent a selected set of genes, arranged according to their genomic coordinates (chromosome number indicated at left). A set of 240 normal mammary epithelial cells is shown on the left for comparison, and epithelial cells from all TNBC cases are shown, clustered separately for each patient. Amplifications (red) or deletions (blue) are inferred by computing, for each gene, a 100-gene moving average expression score, centered at the gene of interest. Prominent subclones defined by shared CNVs in tumors 39 and 81 are indicated by brackets on the top ("clonal"). **e** WES data for four of the six TNBC cases demonstrates high concordance with the CNV calls inferred from the transcriptomes of single cells (**d**). Genomic coordinates are arranged as in **d** from top to bottom, and mean copy number for each region ("CNV mean") is indicated on a continuous scale, with red representing gain and blue representing loss. Accordingly, scanning from left (**d**) to right (**e**) allows for a comparison of inferred CNVs (**d**) and actual CNVs (**e**) for the same regions. **f** Correlation map among the expression profiles of the normal epithelial cells and the TNBC epithelial cells, depicted in the same order from left to right as **d**. Normal cells, as well as malignant clonal subpopulations defined by shared CNVs for tumors 39 and 81 (indicated as "clonal" at top), are correlated. The remaining non-clonal epithelial populations in all tumors show relatively poor correlation, supporting their identity as malignant cells

alluding to particular intratumoral heterogeneity in breast cancers and the existence of subpopulations defined by common states[22,23].

We then sought to further define malignant cells among the epithelial population by identifying large-scale CNVs inferred from single-cell gene expression profiles using a previously described approach[13,21]. For normalization purposes, we used single-cell gene expression data from normal mammary epithelia[23]. Previous single-cell genomic studies established that TNBCs exhibit highly variable patterns of CNVs, driven in part by punctuated tumor genome evolution[7,12]. Consequently, although some TNBCs demonstrate subclones characterized by large CNVs, the subclones within many tumors exhibit relatively small CNVs that are beyond the resolution of inference by transcriptome data[12]. As predicted, we found that subclonal large-scale CNVs were evident in some (particularly tumors 39 and 81) but not all tumors, and no tumors showed a clonal pattern of CNVs shared by all cells (Fig. 2d). We then validated these findings by bulk whole-exome sequencing (WES, Fig. 2e; Supplementary

Fig. 3), which demonstrated high concordance between CNVs inferred by single-cell transcriptomic data and CNVs demonstrated in the genomic data (Fig. 2d, e). For example, gains in chromosome 1 were evident by both approaches in tumors 39 and 81; distinct loss in chromosome 5 and gain in chromosome 12 were verified in tumor 39, while gain in chromosome 8 was evident in tumor 81. Additionally, tumor 89, which demonstrated an absence of inferred clonal or subclonal gains or losses, also showed very few copy number alterations by WES. We also found that the size of the primary tumor at diagnosis was associated with the presence of a dominant subclone, evident in both the scRNAseq-inferred CNVs and the bulk WES. For instance, tumor 39 (9.5 cm tumor) exhibited a dominant subclonal population, whereas tumor 89 (1.5 cm tumor) showed little evidence for such a population (Fig. 2d, e). Despite the small number of tumors, this finding extends our prior single-cell genomic analysis documenting stable aneuploid rearrangements, and suggests that such alterations govern transcription during clonal expansion in breast cancer[7].

Since the absence of clonal CNVs in some tumors precluded the assignment of each epithelial cell as benign or malignant, we next investigated transcriptional heterogeneity within and between the tumor epithelial populations compared to the heterogeneity among normal primary mammary epithelial cells[19] (Fig. 2f). Based on the findings of recent single-cell analyses in other tumor types, we reasoned that non-malignant epithelial cells would be highly concordant, as would tumor epithelial populations defined by subclonal CNVs, whereas the remaining malignant cells would likely be heterogenous and thus non-concordant[13]. As anticipated, the normal epithelial cells showed a reasonably high degree of concordance (mean Spearman correlation 0.38), as did the subgroups of cells from patients 39 and 81 defined by subclonal CNVs (Spearman correlations 0.45 for tumor 39 and 0.41 for tumor 81), which were also identified as distinct clusters of epithelial cells (Fig. 2c; Supplementary Fig. 4). In contrast, the remaining tumor epithelial cells generally showed weaker concordance both within and between tumors, supporting their heterogeneity and therefore their likely identity as malignant cells (Spearman correlations: 0.25 for patient 89, 0.26 for 84, 0.30 for 58 and 126) (Fig. 2f). Taken together, these data demonstrate that epithelial cells within these tumors are characterized either by expression patterns reflecting subclonal CNVs, or by a lack of CNVs and a corresponding high degree of intercellular heterogeneity. These findings and the presence of cycling cells almost exclusively within the epithelial compartment collectively support the hypothesis that the majority of the identified epithelial cells are malignant cells.

**Shared malignant subpopulations reflect diverse phenotypes.** As our clustering analysis demonstrated subgroups of epithelial cells sharing common transcriptional profiles but derived from multiple tumors (Fig. 2c), we next sought to reveal the common biology of these groups via clustering of all epithelial cells while excluding patient-specific effects through linear regression. Using this approach, we identified five clusters of cells, of which one (cluster 2) was represented by a substantial proportion of cells in all tumors, and another (cluster 3) was present in five of six tumors (Fig. 3a, b). Cluster 4 was most prominent in tumor 81, the one tumor that lacked cluster 3 cells, while clusters 1 and 5 represented <70 cells in total and were present only in tumors 84 and 89 (Fig. 3b). Importantly, the majority of cells in cluster 2 (55%) contained large-scale CNVs, thereby identifying this cluster as consisting of malignant cells (Supplementary Fig. 5). Accordingly, cluster 2 contained the highest proportion of high cycling cells (40%), while both clusters 3 and 4 also contained significant high cycling populations (11 and 13% high cycling cells, respectively), and clusters 1 and 5 did not (1 high cycling cell/cluster) (Supplementary Fig. 6). Taken together with the analyses in Fig. 2, these findings support the malignant identity of clusters 2, 3, and 4, while the small number of cells in clusters 1 and 5 can less confidently be identified as malignant cells due to their less proliferative phenotype.

We then investigated the malignant cell clusters and the individual tumors for enrichment of gene expression signatures derived from bulk RNA-seq of established basal, luminal progenitor (LP), and mature luminal (ML) cells from the normal mammary gland (Fig. 3b, c; Supplementary Fig. 7). Clusters 2 and 4, which comprise the majority of malignant cells, were most highly associated with the LP signature, in keeping with data supporting the LP as the cell of origin for breast cancers[24,25]. Cluster 3, in contrast, was clearly distinguished by the ML cell signature. Thus, we found that distinct subpopulations of cycling cells bearing malignant phenotypes are characterized by expression profiles reflecting a spectrum of differentiation along the luminal epithelial lineage.

We next investigated the cluster subpopulations using a set of gene expression signatures derived from unsupervised analysis of bulk breast tumors. These tumor-specific signatures included the TNBCtype-4 signatures, which define certain clinical and biological features of four TNBC subtypes[26,27], and the Intrinsic Basal signature, which was established by comparison across breast cancer subtypes (ER+, HER2+, and TNBC)[28]. Intrinsic Basal tumors represent the majority of TNBCs; they overlap with multiple TNBCtype subtypes, and are associated with increased clonal heterogeneity compared to non-Intrinsic Basal TNBCs (Fig. 3d; Supplementary Figs. 8, 9)[3,28]. We found that the cluster 2 subpopulation was most strongly associated with a single TNBCtype signature known as "Basal Like 1", which is enriched for cell cycle and DNA repair genes, in agreement with the high proportion of cycling cells present in this cluster (Fig. 3e; Supplementary Fig. 8). This cluster was also the most highly enriched for the Intrinsic Basal signature (Fig. 3d; Supplementary Fig. 9). Cluster 4 was also enriched for the Basal-Like 1 signature, while cluster 3 was most highly enriched for the TNBCtype "Luminal Androgen Receptor" signature, concordant with enrichment in this cluster of the differentiated luminal (ML) signature (Fig. 3b–e; Supplementary Fig. 8)[29]. Therefore, the cluster subpopulations are distinct in their expression of established TNBC phenotypes. Of note, in Fig. 3a two potential "subclusters" of clusters 2 and 3 are apparent. Applying multiple signatures to these subpopulations showed they are not consistently different from their respective main clusters, although the cluster 3 subcluster did show more mature luminal character than cluster 3 as a whole (Supplementary Fig. 10).

Performing this analysis for each individual tumor, we found that tumor 81 was comprised predominantly of Basal-Like 1 cells corresponding to LP-like clusters 2 and 4, and had the highest proportion of cycling cells (31%). This finding agrees with the relative homogeneity of this tumor documented by CNV and intercellular heterogeneity analyses (Fig. 2d–f; Supplementary Figs. 11, 12). Tumors 84 and 89, in contrast, included Basal-Like 1/LP cells but also a substantial subpopulation of more differentiated luminal-like cells expressing the luminal androgen receptor and ML signatures. Overall, in the majority of tumors, multiple TNBCtype subtypes were identified (Fig. 3g). Thus, while the most prevalent malignant population within TNBCs corresponds to cells with characteristics of proliferative luminal progenitors, we revealed distinct cell subsets within each tumor, demonstrating the presence of cells with discrete epithelial differentiation status and diverse malignant transcriptional phenotypes. These findings parallel data from single-cell analysis of glioblastoma that demonstrated intra-tumor heterogeneity of gene expression subtypes[21].

**A TNBC subpopulation generates a clinically relevant signature.** Since single-cell analysis may provide enhanced power to reveal tumor cell subpopulations driving poor clinical outcomes, we analyzed the malignant cluster subpopulations for enrichment of distinct gene expression signatures related to aggressive clinical behavior[30–33] (Fig. 4a; Supplementary Figs. 13–15). These include a 70-gene prognostic signature that was initially derived from an analysis of genes differentially expressed between primary tumors of patients who did versus did not experience metastatic relapse[30,31]. A second signature (49-gene metastatic burden signature) distinguishes high versus low metastatic burden conferred by single circulating metastatic cells identified in patient-derived murine xenograft models of TNBC[32]. The third signature (a 354-gene residual tumor signature) was obtained from genes enriched in the residual viable tumor population of patients undergoing pre-operative chemotherapy for treatment of their primary breast cancer[33]. Notably, the overlap among

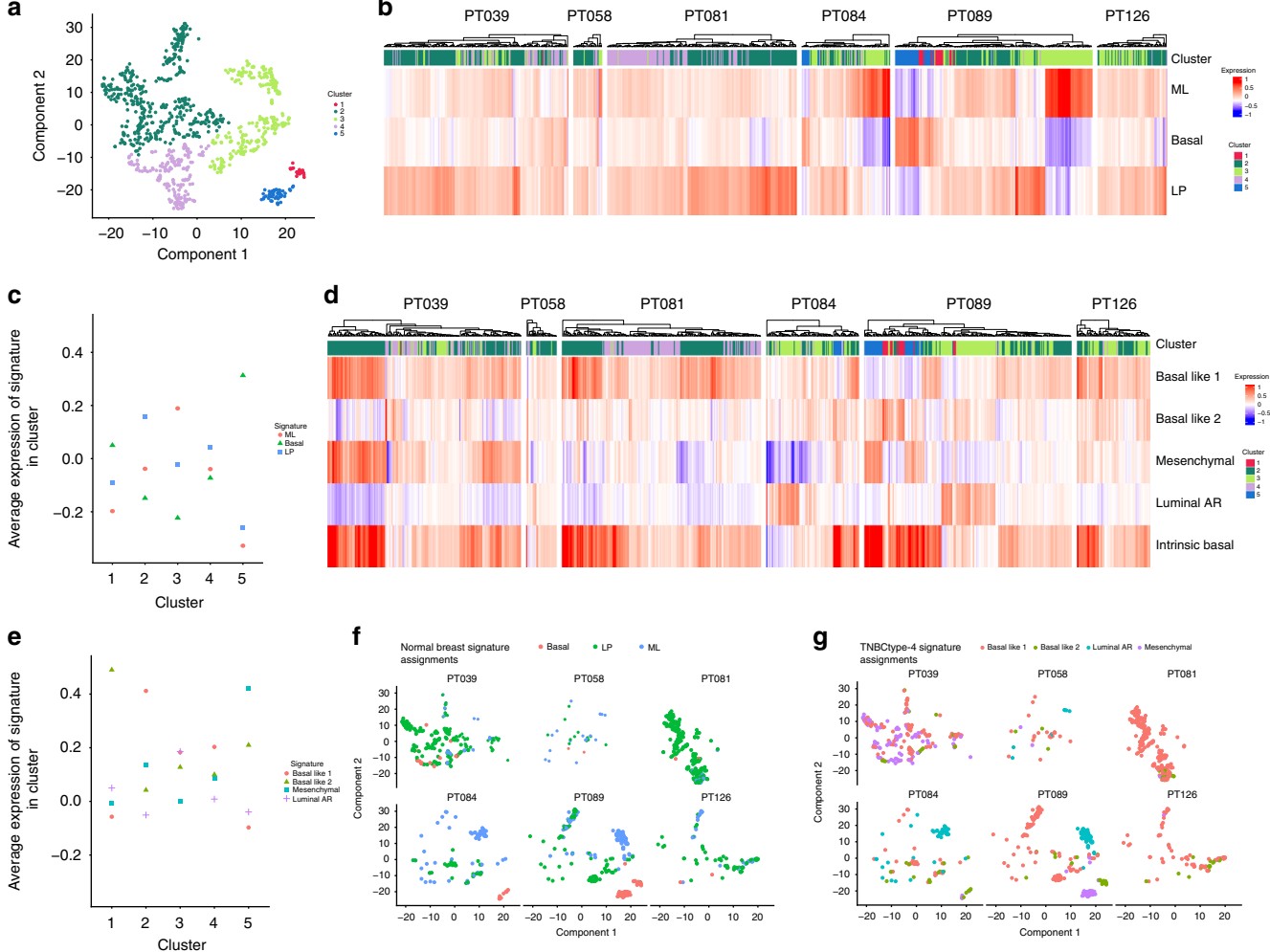

**Fig. 3** Subpopulations of malignant epithelial cells share common expression profiles. **a** t-SNE plot of epithelial cells showing the five identified clusters. Patient-specific effects have been excluded through linear regression analysis. **b** Heatmap depicting the cluster assigned to each cell (top) and the corresponding expression of three normal breast epithelial subtypes signatures: ML (mature luminal), basal, and LP (luminal progenitor). **c** Average expression of each of the three normal breast epithelial subtype signatures in the epithelial clusters. Clusters 2 and 4 most strongly express the LP signature, while cluster 3 most highly expresses the ML signature. **d** Heatmap depicting the cluster assigned to each cell (top) and the corresponding expression of the four TNBCtype-4 subtype signatures and the Intrinsic Basal signature. **e** Average expression of each of the four TNBCtype-4 subtype signatures in the epithelial clusters. Clusters 2 and 4 most strongly express the proliferative Basal-Like 1 signature, while cluster 3 prominently expresses this signature and the luminal AR signature. **f** Assignment of each TNBC epithelial cell to a single normal breast epithelial subtype signature, depending on the difference between its average expression of the upregulated genes characterizing the signature, and the average expression of the downregulated genes. The plurality of cells in all tumors is LP-like, except for tumor 84, which is predominantly comprised of ML-like cells. **g** Assignments of each TNBC epithelial cell to a single TNBCtype-4 subtype signature depending on the difference between its average expression of the upregulated genes characterizing the signature, and the average expression of the downregulated genes. Multiple TNBCtype-4 subtypes are expressed among the cells of each tumor

these signatures was small, and no single gene was present in all three signatures (Supplementary Fig. 16). Remarkably, despite their diverse derivations and small degree of overlap, all three aggressive disease signatures were most highly enriched in the cluster 2 subpopulation (Fig. 4a, b).

To further investigate the biology of cluster 2 cells, we identified the subset of genes differentially expressed between cells in this cluster and all other epithelial cells. We found that these cluster 2-selective genes were significantly associated with genomic copy number gains identified by WES analysis in the three tumors that demonstrated such gains (39, 81, and 84) (Supplementary Table 2). In contrast, the gene sets characterizing the other two malignant clusters (3 and 4) did not demonstrate significant associations between their genomic and transcriptomic profiles. These findings collectively suggest that cluster 2 cells may

drive tumor progression and thereby confer poor outcomes to patients whose tumors contain significant numbers of such cells.

To test this hypothesis, we next derived a unique signature consisting of the top most significantly differentially expressed genes in cluster 2. We applied this signature to a large publicly available data set containing bulk RNA-seq profiles of primary TNBC linked to long-term patient outcomes, the METABRIC cohort[34]. We observed a statistically significant association between tumors with high expression of the cluster 2 signature and shortened overall survival (Fig. 4c). In contrast, none of the three original aggressive disease signatures that were enriched in cluster 2 were themselves predictive of outcomes in this patient cohort (Fig. 4b, c). Furthermore, none of the signatures defining clusters 1, 3, 4, or 5 were associated with clinical outcomes in this cohort (Supplementary Fig. 17). Similarly, the intrinsic Basal

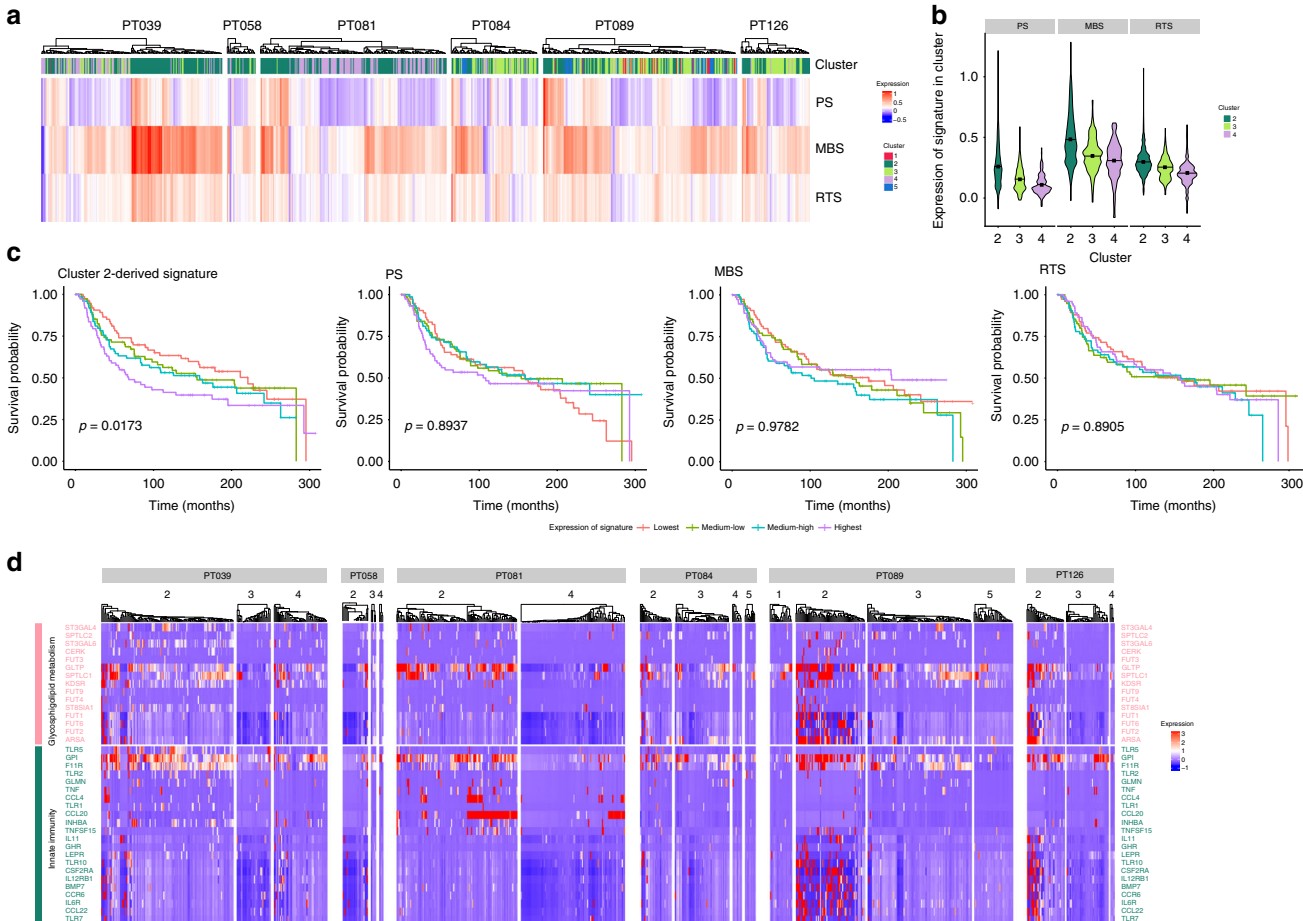

**Fig. 4** A cluster 2 subpopulation signature predicts poor patient outcomes and reflects glycosphingolipid and innate immunity pathways. **a** Heatmap depicting the cluster assigned to each cell (top) and the corresponding expression of three signatures related to aggressive disease behavior: 70-gene prognostic signature (PS), 49-gene metastatic burden signature (MBS), and 354-gene residual tumor signature (RTS) (rows). **b** Violin plots representing the distribution of expression among cells within the indicated clusters of the three signatures related to aggressive disease behavior. Black squares represent average expression of each signature among the cells of the corresponding cluster. **c** Kaplan–Meier survival curves for TNBC patients in the METABRIC cohort by expression quartiles of the cluster 2-derived gene signature (left). Higher expression of the signature is significantly associated with worse patient outcome (log-rank test, $p = 0.0173$). The other three signatures related to aggressive disease behavior are not predictive of survival (right). Separation on quartiles is for visualization purposes. **d** Heatmap demonstrating expression of genes in the glycosphingolipid metabolism and innate immunity pathways in the epithelial clusters (indicated at top) across all patients. Cluster 2 is significantly enriched for expression of genes in both pathways

TNBC signature was not associated with clinical outcomes in this cohort (Supplementary Fig. 18). Collectively, these findings reveal the ability of single-cell analysis to unveil clinically relevant cell states that have not been uncovered through bulk tumor analysis.

**Metabolism and immunity programs characterize the poor prognosis subpopulation.** We then sought to determine the functional programs underlying the cluster 2 subpopulation by investigating the pathways enriched among the genes differentially expressed in cluster 2 compared to the other epithelial cells[35]. We found that the most enriched pathways were associated with glycosphingolipid biosynthesis and lysosomal turnover, which impinge on cytokine pathways of the innate immune system that were also enriched (Fig. 4d; Supplementary Table 3). Glycosphingolipids have recently been implicated as mediators of numerous tumor-promoting properties in breast cancer, including altered growth factor signaling, EMT, and stem-like behavior[36–38]. Multiple key genes in this pathway are selectively expressed in the cluster 2 subpopulation, including the glycolipid transfer protein gene *GLTP* and the key sphingolipid biosynthesis

subunit gene *SPTLC1*. Glycosphingolipids are also recognized as important modulators of both the innate and adaptive immune systems, and have been linked to inflammation-associated carcinogenesis in multiple epithelial tissues[39]. Indeed IHC staining confirmed expression of *SPTLC1* in a subpopulation of cells in all six TNBCs, and in addition we observed high levels of the sphingosine-1-phosphate receptor *S1PR1*, which functions in a reciprocal feedback loop to activate *STAT3* in multiple cancer contexts[40], upon IHC staining in all tumors (Supplementary Figs. 19, 20). Another notable cluster 2-selective gene related to innate immunity in the epithelium is *GPI/AMF* (glucose-6-phosphate isomerase/autocrine motility factor), a tumor-secreted cytokine implicated in EMT, migration, and metastasis (Fig. 4d)[41,42]. Additionally, expression of the epithelial tight junction assembly factor gene *F11R* has been linked to breast cancer progression and patient survival[43]. Deregulation of barrier factors such as *F11R* can induce established pro-tumorigenic cytokines, and indeed a subset of these, including *CCL20* and *CCL22*[44,45], is highly expressed selectively in the cluster 2 subpopulation.

Finally, we determined the impact of the glycosphingolipid pathway expression itself on clinical outcomes in order to validate

its relevance for the cluster 2 subpopulation specifically and to TNBC in general. We found that a gene signature representative of glycosphingolipid metabolism[46] (Supplementary Table 4) was predictive of overall survival for TNBC patients in the METABRIC cohort, with progressively higher expression significantly associated with increasingly worse overall survival in this TNBC cohort (Supplementary Fig. 21). Collectively, these findings reveal an unanticipated subpopulation of TNBC cells whose transcriptome reflects genomic evolution and whose distinct biology confers poor clinical outcomes for TNBC patients.

## Discussion

Here we provide fundamental new insights into the subclonal biology of TNBC through single-cell RNA-sequencing of >1500 cells obtained from six fresh tumors. We characterized the extent of diversity among TNBC cells with regard to their expression of normal breast and established TNBC signatures, identifying a subpopulation of cells, shared among tumors, whose properties drive poor outcomes. These findings support accumulating evidence that intra-tumor heterogeneity is central to the clinical behavior of this disease. For example, prior studies have documented that TNBCs exhibit ongoing mutational diversification, giving rise to genomic heterogeneity that can be inferred through deep sequencing and confers worse clinical outcomes[3,7]. Clinically, a failure to achieve pathologic complete response following pre-operative chemotherapy is associated with high relapse rates, consistent with evidence we provide that a minor refractory subpopulation of cells determines patient outcomes.

Our single-cell transcriptome analysis reveals that tumors are comprised of variable proportions of malignant, stromal, and immune components, and we showed that proliferating cells are confined almost exclusively to the malignant cell compartment. Concordant with recent studies reporting single-cell analysis of other tumor types including glioblastoma and melanoma, we found that normal cells cluster together by cell type (not by patient) upon unsupervised analysis[13,21]. The malignant cells in these other tumor types primarily formed tumor-specific clusters. In contrast, we showed that TNBCs are comprised only partly of tumor-specific clusters, which correspond to subpopulations defined by large clonal CNVs that are lacking in some tumors. In addition, we observed subpopulations of cells from multiple tumors that exhibit malignant characteristics and yet cluster together rather than with other cells from the same patient. Taken together, these findings suggest that subclonal diversification can give rise to tumor-specific cell populations, but also imply that TNBCs are characterized by subpopulations with shared biological properties across patients.

We further determined that these shared malignant subpopulations express distinct expression profiles resembling the normal epithelial cell types and previously defined TNBC subtypes. The major cell subtype present in most TNBCs we analyzed was a highly proliferative group most closely related to the normal LP cell, in keeping with the emerging view that LPs represent the cell of origin for many breast cancers[24,25]. The most prevalent tumor-derived (TNBCtype) signature among all cells was the Basal-Like 1 proliferative signature, which corresponds largely to these proliferating LP-like cells. However, most tumors also contain cells with an intermediate proliferative index bearing the more differentiated ML signature and corresponding to the Luminal Androgen Receptor TNBCtype subtype. Our finding that each tumor was comprised of multiple TNBCtype and normal cell subtypes implies that the dominant signature reflected in bulk analysis may be a poor representation of the functional heterogeneity within and between tumors. Significantly, this observation

may explain in part why such gene expression signatures have largely failed as useful clinical diagnostics for TNBC. In contrast, in hormone receptor-positive breast cancer, which exhibits substantially less intra-tumor heterogeneity, gene expression signatures are now routinely integrated into clinical decision-making[47].

Given the established intratumoral heterogeneity of TNBC, it is perhaps not surprising that established signatures derived from bulk normal cells and TNBCs did not reveal a distinct biology of the shared subpopulations we identified. Thus, we interrogated these cluster subpopulations for their expression of diverse signatures related to treatment resistance and metastasis. This analysis pointed to a single malignant subpopulation (cluster 2), present in each tumor we analyzed, that was most highly enriched for each of these aggressive disease phenotypes. Notably, the genes defining the cluster 2 subpopulation (but not other subpopulations) were significantly associated with subclonal genomic copy number gains, underscoring the role of genomic evolution in driving a poor prognosis transcriptional state. Strikingly, the expression of differentially expressed genes specific to cluster 2 was predictive of outcomes in TNBC, while the previously defined signatures related to aggressive tumor behavior, as well as those associated with the other cluster subpopulations, were not. We then revealed the biological pathways associated with these cluster 2 cells, demonstrating enrichment of genes involved in glycosphingolipid/lysosome function, and innate immune sensing and inflammation. Emerging data point to an underappreciated role for glycosphingolipids in multiple tumor phenotypes, while the role of inflammatory responses as a tumor promoter is well-established[48,49]. Finally, we demonstrated that a glycosphingolipid pathway signature itself was a significant predictor of outcomes for TNBC patients.

Given their relevance to clinical outcomes, our findings may form the basis for the future development of patient and tumor-specific markers that could more accurately identify refractory subpopulations at the time of diagnosis, and thereby predict treatment resistance and prognosis in TNBC. Additionally, the specific pathways we implicate include numerous potentially attractive therapeutic targets, including S1PR1, which is highly expressed in these tumors[50–52]. Undoubtedly, future studies will uncover additional features and cell subpopulations that govern tumor behavior, particularly with respect to non-malignant compartments. By unveiling clinically relevant states and their relationship to genomic evolution at the level of individual cells, this study substantiates the far-reaching promise of single-cell analysis in TNBC and other cancer types characterized by extensive intra-tumor heterogeneity.

## Methods

**Human tumor specimens**. Fresh tumors from TNBC specimens (Supplementary Table 1) were collected at Massachusetts General Hospital with approval by the Dana Farber/Harvard Cancer Center Institutional Review Board (93-085), and signed informed consent was obtained from all patients. Five of six patients underwent genetic testing, revealing that patient 84 was a BRCA2 mutation carrier and the others lacked BRCA1/2 mutation, while patient 58 did not undergo such testing. Tumor tissues were mechanically and enzymatically dissociated using tumor dissociation kit (Miltenyi Biotec). Single-cell suspensions were collected after removing large pieces of debris using a 40-μm cell strainer.

**Flow cytometry and sorting**. Tumor cells were blocked in 3% FBS in Hanks buffered saline solution, and then stained first with CD45-Vioblue direct conjugate antibody (130-092-880, Miltenyi Biotec, Bergisch Gladbach, Germany). Cells were washed and then stained for viability (Calcein AM, TO-PRO-3, Life Technologies, Carlsbad, CA, USA). FACS was performed on FACSAria Fusion (Becton Dickinson, Franklin Lakes, NJ, USA). Strict singlets were selected by using standard criteria for forward scatter height versus area. Viable cells were identified by staining positive with Calcein (FITC) and negative for TO-PRO-3 (APC). Single cells were sorted into 96-well plates containing 10 μl TCL buffer (Qiagen, Hilden,

Germany) + 1% β-mercaptoethanol. Plates were spun briefly, snap-frozen on dry ice immediately, and then stored at −80 °C until further processing.

**cDNA synthesis, library construction, and sequencing**. Smart-seq2 was performed on single sorted cells[53], with the following modifications: RNA was cleaned up using Agencourt RNAClean XP beads (Beckman Coulter, Pasadena, CA, USA). Reverse transcription was carried out using oligo-dT primers, Maxima reverse transcriptase and locked TSO oligonucleotide prior to PCR amplication with KAPA HiFi HotStart ReadyMix (Kapa Biosystems). Subsequently, Agencourt AMPure XP bead (Beckman Coulter, Pasadena, CA, USA) purification was applied. Full-length cDNA libraries were barcoded using the Nextera XT Tagmentation protocol (Illumina, San Diego, CA, USA). Libraries from 96 pooled cells were sequenced as 38 bp paired end on NextSeq 500 (Illumina, San Diego, CA, USA).

**Bioinformatics analysis**. FASTQ files were quantified to transcript per million (TPM) expression values with RSEM[54] with default parameters. Quality control was performed by removing both low quality cells, as well as genes with too low expression. We identified low quality cells by (i) small library size, (ii) few expressed genes, and (iii) low total amount of mRNA. All cells that were at least 4 median absolute deviations (MADs) below the median for any of these three metrics were removed from downstream analyses[55] (Supplementary Fig. 22). For each of the six patients, we identified the genes that were not expressed ($\log_2$ (TPM + 1) < 0.1) in at least 95% of cells for that respective patient, and removed the intersection of these six sets from the set of all genes. A total of 13280 genes remained after filtering. To normalize single-cell RNA-seq data, a three step strategy was employed: (1) transform the TPM values into relative counts with the Census algorithm (function relative2abs from the R package monocle[56]); (2) normalize the Census counts with the deconvolution strategy implemented in the R package scran[57]; (3) remove additional sources of unwanted variation in the scran-normalized Census counts with RUVSeq[58](RUVg) (Supplementary Fig. 23). After removing all cells with size factors equal to 0, 1189 cells remained for downstream analyses (Supplementary Table 5). This normalization strategy strongly reduced the impact of many known sources of technical variability and confounding factors from the expression data (Supplementary Figs. 24–30 and Supplementary Table 6). For more details, see Supplementary Methods.

Identifying cell types: We employed a two-step combination approach to identify the different cell types that tumors consist of: (1) literature-based list of specific expression markers previously established to define cell types (Supplementary Table 7); (2) clustering (Supplementary Tables 8 and 9; see Supplementary Methods).

Identifying cycling cells: To identify cycling cells, scores for the G1-S and G2-M phases of the cell cycle were computed by averaging the expression of a set of relevant genes[13]. Cycling cells were defined to be the ones with high G1-S score or G2-M score, and non-cycling cells were the ones with low G1-S and G2-M scores[13]. Data-derived thresholds of 2 MADs above the median were used to decide whether a score is high or low (see Supplementary Methods).

**Identifying copy number alterations from scRNA-seq data**. The expression profiles were normalized by subtracting, from the expression of each cell, the average expression of 240 normal epithelial cells profiled in a different study[23]. 20,337 transcripts were common between our data set and the data set profiling the normal epithelial cells. Expression was quantified as $\log_2$ (TPM + 1)/10, and all genes with average expression across all cells <0.1 were removed. This amounted to keeping 4673 transcripts. Genes were ordered by their chromosomal location after removing the average expression of the 240 normal cells. All expressions greater than 3 and lower than −3 were leveled. The copy number value of each gene was defined as the sliding average value with a window size of 100 and centered at the gene of interest. Finally, for each gene, the resulting copy number values were centered across all cells (see Supplementary Methods).

**Clustering of epithelial cells**. The 886 epithelial cells were clustered using the algorithm developed in Monocle and regressing out the patient effect. The number of clusters was automatically chosen by Monocle, implementing a density-based approach[59]. Five epithelial clusters were identified as follows: cluster 1 (22 cells); cluster 2 (398 cells); cluster 3 (231 cells); cluster 4 (170 cells); cluster 5 (47 cells) (see Supplementary Methods).

**Gene expression signatures**. The expression of each cell under the normal breast and TNBCtype-4 signatures was computed by subtracting the mean expression of the downregulated genes from the mean expression of the upregulated genes. Each cell was assigned to the signature for which it had highest expression. For the prognostic signature, metastatic burden signature, residual tumor signature and intrinsic basal signature, the expression of each cell under each signature was computed by averaging the mean expression of its genes. Expression heatmaps were plotted using the ComplexHeatmap R package[60] (see Supplementary Methods).

**Survival analysis**. Survival analyses were performed using Cox proportional hazards regression models, and p-values were obtained from log-rank tests (see Supplementary Methods).

**Whole-exome sequencing**. Library construction, data processing, and copy number profiling of the exome data were performed as described in Supplementary Methods.

**Immunohistochemistry**. Five micron sections were cut and stained for anti-Ki67 (M7240, Agilent Dako, Santa Clara, CA, USA), anti-SPTLC1 (HPA010860, Atlas Antibodies, Voltavägen, Bromma, Sweden), and anti-S1PR1 (ab11424, Abcam, Cambridge, MA, USA) using standard protocol.

**Code availability**. All computational analyses were performed in R (version 3.4.3). The code used for these analyses is available at naseq.

## Data availability

All data supporting the findings of this study are available within the article and its supplementary information files or upon request. The scRNAseq and WES data have been deposited in the Gene Expression Omnibus (GEO) database under accession code GSE118390.

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

## Acknowledgements

We thank the Dana-Farber Center for Genome Discovery for library processing for exome sequencing, MGH Histopathology Research Core for immunohistochemistry experiments, Elena Zarcaro for patient consent and database management, Sowmya Iyer, Ryanne Boursiquot, Anuraag S. Parikh for consultation, and S. Vinod Saladi for critical review of the manuscript. This work was funded by Rosanne's Rush for Research and the Tracey Davis Memorial Fund. B.E.B and F.M. are supported by Ludwig Cancer Research. M.K. is supported by the Susan G. Komen Foundation and the Terri Brodeur Breast Cancer Foundation. S.C. is supported by the Swiss National Science Foundation, project number P2EZP2_175139. Support from the Dana-Farber Physical Sciences Oncology Center (NIH U54CA193461, to F.M.) is gratefully acknowledged.

## Author contributions

M.K., B.E.B. and L.W.E. conceived and designed the study. M.C.S. contributed patient samples. S.C., F.M. and M.K. designed the analysis and developed methods. S.C., F.M., M.K. and L.W.E. performed the data analysis. M.K., S.M.G., A.P.P., R.M. and C.C.L. performed experiments. S.C. contributed data and analysis tools. L.W.E., S.C., M.K. and F.M. wrote the manuscript.

## Additional information

**Competing interests:** The authors declare no competing interests.

