## [Peer Review File · Nature Communications]

Reviewers' comments:

Reviewer #1 (Remarks to the Author):

In this manuscript the authors perform single-cell RNA sequencing on over 1500 cells from six triple-negative breast cancer (TNBC) patients to uncover the intracellular heterogeneity known to be part of the disease. They demonstrate that intracellular heterogeneity correlates with clonality of inferred genomic copy number changes and gene expression profiles identify distinct subgroups of malignant cells shared by multiple tumors with their compositions varying by tumor. While this was a thorough analysis in a larger population of TNBC cells, the conclusions largely reinforce those from a previous Nature Communications publication by Chung et. al. (PMID: 28474673). The authors also identify a cluster that is associated with poor outcome and characterized by enrichment in glycosphingolipid biosynthesis and lysosomal turnover pathways and innate immunity.

Specific Comments

1. The cycling status in Figure 1B is labeled "cycling" and "non-cycling". This is not appropriately labeled as the non-cycling cells could still be cycling at a much slower rate. Please modify to "high cycling" and "low cycling" that more accurately describes the cell state and matches the labels in Figure 1E.

2. TNBC subtyping was performed using an unconventional method by subtracting the mean expression of the downregulated genes from the mean expression of the upregulated genes. Please use the TNBCtype software (<http://cbc.mc.vanderbilt.edu/tnbc/>) and use similar methods as Chung W. et al. (PMID: 28474673) that include removal of genes not expressed in any single TNBC cell and upload without centering to avoid false ER+ calling as the result of zero ESR1 expression in most cells.

3. While the investigators have chosen an unbiased method to automatically choose the number of clusters (Monocle) and found five transcriptional clusters (Fig 3A). There appears to be at least two additional clusters within clusters 2 and 3 that appear obvious by eye (See attached review figure). The authors should consider evaluating these two clusters individually as their biologies may differ.

4. In Figure 3A, Cluster 3 appears to be composed of two clusters and half of cluster 3 appears to be composed of a more mature luminal and LAR TNBC subtype. Does the LAR subtype or mature luminal signature enrich in the top or bottom cluster and should these be two separate clusters? There is also a similar separation in cluster 2, with two groups in Figure 3A. Perhaps performing the same t-SNE plots as in 3A, but labeling with TNBC subtype or normal breast signatures could be helpful in distinguishing these subsets in addition to the clusters suggested by Monocle.

5. The authors have chosen not to evaluate clusters 1 and 5 due to their limited number of cells and less proliferative phenotype. However, these clusters may be biologically important and reflect a small population of "cancer stem cells". Furthermore cluster 1 and 5 appear to be the only true basal cells using normal lineage signatures and only present in pt89 after removal of stromal cells. These two clusters appear to be correlated to BL2 and M (or more likely MSL) TNBC subtypes. They also express myoepithelial markers including ACTA2 and TAGLN. Please discuss more about this lineage difference and whether these cells are likely from a different (basal cell) origin. Do they also express other myoepithelial markers (TP63 and MME)? Please add more discussion of these results.

6. In Figure 4C the unique signature of cluster 2 is compared to three previously published poor prognosis signatures using METABRIC. It would be interesting to see how cluster 2 compares to signatures derived from the other single cell clusters in this dataset as well.

7. In Figure S16 there is no control to demonstrate specificity of S1PR1 staining. Furthermore all of the patients stained by IHC displayed high levels of S1PR1 regardless of cell cluster composition. For example pt84 displayed the least amount of cluster 2 derived cells (Fig 3B), but intense staining for S1PR1. Please address this discrepancy.

Reviewer #2 (Remarks to the Author):

In this paper, the authors use single cell RNA sequencing to examine cellular heterogeneity in triple negative breast cancer tumors. Using clustering together with known markers, they were able to parse cells into various types, with the plurality being epithelial. To identify which cells were malignant, they used cell cycle analysis, along with using CNV analysis to find subclones, suggesting that the epithelial cells are the exclusively malignant ones. The putative malignant cells expressed LP markers, supporting their identification as the cell of origin. They then use single cell signatures to try and classify outcomes, showing some association with prognosis.

Overall, I think this is an interesting and timely paper. Single cell analysis in cancer is a area of significant interest, and I thought the combination with CNV analysis was an interesting new addition. The use of single cell signatures as a prognostic has a lot of potential, and I agree with the authors' line of argumentation that prognostics could benefit from analysis of rare subpopulations that are generally glossed over. The data and analysis appear to be of high quality, and the claims are well supported.

I have a few minor comments for improvement:

The CNV results were quite confusing to me, both in the text and in the figure. The graphics in the figure made it very unclear as to what was being correlated with what, and the numerical quantifications were not particularly intuitive (what is expression score? CNV mean?, etc.).

Perhaps I'm being dense, but I didn't quite understand the rationale behind the statement "We reasoned that non-malignant epithelial cells would be highly concordant, as would tumor epithelial populations defined by subclonal CNVs, whereas the remaining malignant cells would likely be heterogenous and thus non-concordant." Why would this a priori be reasonable? Why would malignant cells not be more clonally related?

In the textual description of Fig. 4, it would be helpful to have some sort of estimate of the increased predictive power beyond just a statement of statistical significance. How much better does this do than, say, just the straight bulk expression profile?

Reply to Reviewers

We would like to thank both reviewers for their thorough reading of our manuscript and for their constructive comments. We were particularly pleased that the reviewers felt that “this was a thorough analysis in a larger population of TNBC cells” (Rev 1); “an interesting and timely paper” and that “The data and analysis appear to be of high quality, and the claims are well supported.” (Rev 2). We have now addressed all points raised in a revised version of our paper.

Please note that for clarity, all Reviewer comments appear in Times Italic font, and all responses in Arial font.

Reviewer #1 (Remarks to the Author):

In this manuscript the authors perform single-cell RNA sequencing on over 1500 cells from six triple-negative breast cancer (TNBC) patients to uncover the intracellular heterogeneity known to be part of the disease. They demonstrate that intracellular heterogeneity correlates with clonality of inferred genomic copy number changes and gene expression profiles identify distinct subgroups of malignant cells shared by multiple tumors with their compositions varying by tumor. While this was a thorough analysis in a larger population of TNBC cells, the conclusions largely reinforce those from a previous Nature Communications publication by Chung et. al. (PMID: 28474673).

The authors also identify a cluster that is associated with poor outcome and characterized by enrichment in glycosphingolipid biosynthesis and lysosomal turnover pathways and innate immunity.

Reply: We agree with the reviewer that the Chung et al paper (our Ref. #22) was an important contribution to the field. We note that our manuscript involves analysis of a substantially larger number of cells, exclusively from TNBC, and therefore provides a more in-depth analysis of this challenging breast cancer subtype. In addition, we demonstrate that a shared tumor cell subpopulation (cluster 2) is associated with sub-clonal copy number variations (Supplementary Fig. 5), suggesting that this group of cells represents an expanding subpopulation within the tumors. Most importantly, we show that this cluster is enriched for signatures of aggressive disease and bad outcomes (Fig. 4b), and that the signature defining cluster 2 is associated with clinical outcomes in TNBC (Fig. 4c). To our knowledge, this is the first demonstration of patient outcomes associated with a subclonal population defined by single-cell RNAseq within TNBC.

Specific Comments

1. The cycling status in Figure 1B is labeled “cycling” and “non-cycling”. This is not appropriately labeled as the non-cycling cells could still be cycling at a much slower rate. Please modify to “high cycling” and “low cycling” that more accurately describes the cell state and matches the labels in Figure 1E.

Reply: Thank you for this suggestion; we have changed the legends to Figures 1b, d, and e, and Supplementary Figure 2 accordingly.

2. TNBC subtyping was performed using an unconventional method by subtracting the mean expression of the downregulated genes from the mean expression of the upregulated genes. Please use the TNBCtype software (<http://cbc.mc.vanderbilt.edu/tnbc/>) and use similar methods as Chung W. et al. (PMID: 28474673) that include removal of genes not expressed in any single TNBC cell and upload without centering to avoid false ER+ calling as the result of zero ESR1 expression in most cells.

Reply: We concur with the reviewer that this would be an interesting comparison between methodologies for TNBCtype analysis. Unfortunately, we have been unable to successfully analyze our data using the online software cited, despite removal of non-expressed genes and uploading without centering as suggested. In each case, multiple error messages are generated. More specifically, among the 868 epithelial cells in our data, only 260 were considered valid to be classified by the online TNBCtype analysis. The remaining cells were labeled as “possibly ER-positive”, defined as cells with more than 75% genes’ expression lower than ESR1’s. We attempted to overcome this issue by more stringent preprocessing, resulting in the removal of more genes with average low expression, and we tried using various low average expression thresholds. However, for low thresholds, the ER-positive issue arose for the exact same number of samples, whereas for larger thresholds, the software did not run because too few of a background list of 2,000 genes were found in the data. As we are unsure what this list consists of, we were unable to enforce those genes to not be removed.

Undoubtedly, the issues preventing us from successfully running the Vanderbilt software directly follow from the differences between bulk and single cell expression data. In contrast to bulk RNA-seq data, single cell expression data is known to be affected by a strong drop-out effect, which makes the exact expression of any single particular gene such as ESR1, not necessarily a robust indicator of the state of the cell. We did contact the authors of the Chung et al paper regarding this analysis, but they were unable to provide any additional information beyond what was stated in their manuscript. Ultimately, TNBCtype analysis of single cell data is likely to differ from bulk analysis regardless of the methodology used. As a whole, we believe that our analyses, together with that of the normal signatures (Fig. 3b), are an informative addition to the manuscript.

3. While the investigators have chosen an unbiased method to automatically choose the number of clusters (Monocle) and found five transcriptional clusters (Fig 3A). There appears to be at least two additional clusters within clusters 2 and 3 that appear obvious by eye (See attached review figure). The authors should consider evaluating these two clusters individually as their biologies may differ.

4. In Figure 3A, Cluster 3 appears to be composed of two clusters and half of cluster 3 appears to be composed of a more mature luminal and LAR TNBC subtype. Does the LAR subtype or mature luminal signature enrich in the top or bottom cluster and should these be two separate clusters? There is also a similar separation in cluster 2, with two groups in Figure 3A. Perhaps performing the same t-SNE plots as in 3A, but labeling with TNBC subtype or normal breast signatures could be helpful in distinguishing these subsets in addition to the clusters suggested by Monocle.

Reply: We would like to thank the reviewer for these important suggestions. We have now analyzed the smaller sub-clusters associated with cluster 2 (at left) and cluster 3 (at right) shown in 3A (Figure R1A, below). The new analyses show that the two sub-clusters are similar to one another in being more (normal) luminal in character than their respective major clusters (Figure R1B). In contrast, these two sub-clusters differ in their TNBCtype signatures: as suggested by the reviewer, the cluster 3 sub-cluster is more luminal AR than either cluster 2 or cluster 3 as a whole, while cluster 2 sub-cluster is more basal-like 1 (Figure R1C). A third pattern is evident when the cells are analyzed according to the prognosis signature, as the two sub-clusters are different from one another, but they largely correspond to their respective main clusters (Figure R1D). Thus, even though these analyses do support some differences between the sub-clusters and the main clusters, the sub-clusters are however not consistently different. In addition, the character/signatures that define them as different (i.e. the normal and TNBCtype signatures) have not been reported as associated with clinical outcomes in TNBC, unlike the signature that defines cluster 2. Collectively, these findings support the biological relevance of our general cluster assignments.

Figure R1, t-SNE plot from Fig. 3A (panel A), with cells analyzed for signatures described in the manuscript, including the normal signatures (B), the TNBCtype signatures (C), and the 70-gene prognosis signature (D). (Please zoom to see legends.)

5. The authors have chosen not to evaluate clusters 1 and 5 due to their limited number of cells and less proliferative phenotype. However, these clusters may be biologically important and reflect a small population of “cancer stem cells”. Furthermore cluster 1 and 5 appear to be the only true basal cells using normal lineage signatures and only present in pt89 after removal of stromal cells. These two clusters appear to be correlated to BL2 and M (or more likely MSL) TNBC subtypes. They also express myoepithelial markers including ACTA2 and TAGLN. Please discuss more about this lineage difference and whether these cells are likely from a different (basal cell) origin. Do they also express other myoepithelial markers (TP63 and MME)? Please add more discussion of these results.

Reply: Thank you for these suggestions. As shown in Fig. 3 b/c (and Fig. R1), clusters 1 and 5 both exhibit (normal) basal character, whereas by TNBCtype, cluster 1 is distinguished as being

more basal-like 2, while cluster 5 is more mesenchymal (Fig.3 d/e). Interestingly, all four of the myoepithelial markers mentioned above are expressed almost exclusively in cluster 5 rather than cluster 1 (Fig. R2), thus defining the former population as most closely associated with the myoepithelial lineage. To investigate whether these cells might represent tumor stem cells that could dictate clinical outcomes, we developed a signature of each cluster, employing the same methodology as we did to generate the predictive signature characterizing cluster 2. That is, a signature comprised of the top most significantly differentially expressed genes between that cluster and all the other epithelial cells. Using these signatures, we carried out clinical outcome analysis in the same way as we had for the cluster 2 signature in Fig. 4c. These findings, shown in Fig. R3 and in the new Sup. Fig. 16, demonstrate the absence of an association between patient survival and the signatures associated with clusters 1, 3, 4 and 5. We have now updated the Results section p.14 and Discussion p. 18 to incorporate these points.

Figure R2, t-SNE plot from Fig. 3A, with cells analyzed for expression of the following genes: ACTA2 (A), TAGLN (B), TP63 (C), and MME (D). (Please zoom to see scale.)

6. In Figure 4C the unique signature of cluster 2 is compared to three previously published poor prognosis signatures using METABRIC. It would be interesting to see how cluster 2 compares to signatures derived from the other single cell clusters in this dataset as well.

Reply: Thank you for this excellent suggestion. We have now included the clinical outcome analysis of the signatures representing the four other clusters, derived in the same manner as was done for cluster 2 (Fig R3 and the new Sup. Fig. 16). This analysis shows that none of the signatures defining the other clusters are associated with clinical outcome in TNBC. These findings further reinforce the significance of our findings regarding the cluster 2 sub-population. We have revised the manuscript accordingly as noted in the previous point.

7. In Figure S16 there is no control to demonstrate specificity of SIPR1 staining. Furthermore

all of the patients stained by IHC displayed high levels of S1PR1 regardless of cell cluster composition. For example pt84 displayed the least amount of cluster 2 derived cells (Fig 3B), but intense staining for S1PR1. Please address this discrepancy.

Reply: We apologize for the lack of clarity in the paper regarding this point. S1PR1 was chosen because it could be a potential therapeutic target in TNBC involved in glycosphingolipid metabolism. We completely agree with the reviewer that it would be valuable to show the protein expression of a cluster 2-selective gene in these tumors. Accordingly, we have stained the tumors for SPTLC1 (serine palmitoyl transferase), which is most highly expressed in cells of cluster 2 compared to other clusters (Fig. 4d). As shown in the new Sup. Fig. 18, we observed expression in subpopulations of tumor cells in each patient in a pattern that recapitulates findings obtained from the scRNAseq analysis, such as a small population of intensely expressing cells seen in patients 81 and 89. This new data is incorporated into the revised manuscript (p.15).

Figure R3, Kaplan-Meier analysis of survival among TNBC patients in the METABRIC cohort of the gene signatures defining cluster 1 (A), cluster 3 (B), cluster 4 (C), and cluster 5 (D). p-values from log-rank tests. No significant associations with survival are observed.

Reviewer #2 (Remarks to the Author):

In this paper, the authors use single cell RNA sequencing to examine cellular heterogeneity in triple negative breast cancer tumors. Using clustering together with known markers, they were able to parse cells into various types, with the plurality being epithelial. To identify which cells were malignant, they used cell cycle analysis, along with using CNV analysis to find subclones, suggesting that the epithelial cells are the exclusively malignant ones. The putative malignant cells expressed LP markers, supporting their identification as the cell of origin. They then use single cell signatures to try and classify outcomes, showing some association with prognosis.

Overall, I think this is an interesting and timely paper. Single cell analysis in cancer is a area of significant interest, and I thought the combination with CNV analysis was an interesting new addition. The use of single cell signatures as a prognostic has a lot of potential, and I agree with the authors' line of argumentation that prognostics could benefit from analysis of rare subpopulations that are generally glossed over. The data and analysis appear to be of high quality, and the claims are well supported.

Reply: We thank the reviewer for this excellent summary of our manuscript.

I have a few minor comments for improvement:

The CNV results were quite confusing to me, both in the text and in the figure. The graphics in the figure made it very unclear as to what was being correlated with what, and the numerical quantifications were not particularly intuitive (what is expression score? CNV mean?, etc.).

Reply: We apologize for this confusion and have now revised the legends to Figures 2d and 2e, as well as the manuscript (p. 8) to clarify the CNV analysis. In brief, “expression score” in 2d refers to a 100-gene moving average of gene expression in a region centered at the gene of interest. “CNV mean” in 2e refers to mean genomic copy number of an indicated genomic region, as obtained from whole-exome sequencing. Identical genomic coordinate maps (chromosome number) are shown at left in 2d and 2e from top to bottom. Accordingly, scanning from left to right, the reader can compare inferred CNVs (d) and actual CNVs (e) for the same chromosomal regions in the same tumor. The reader can see, for example, that inferred gains (red) shown in (d) within chromosomes 1 and 12 in tumor 39 are indeed present upon bulk genomic sequencing analysis (e). Additionally, in 2f, the cells are arranged in the same order from left to right as in 2d, allowing the reader to compare intercellular heterogeneity in subpopulations that either contain or lack recognizable CNVs. As noted, this analysis shows that the subpopulation in tumor 39 that contains the chromosomes 1 or 12 gains (identified as “clonal” by brackets at the top in 2d and 2f) is less heterogenous than the remaining (“nonclonal”) cell population in this tumor.

Perhaps I'm being dense, but I didn't quite understand the rationale behind the statement “We reasoned that non-malignant epithelial cells would be highly concordant, as would tumor epithelial populations defined by subclonal CNVs, whereas the remaining malignant cells would likely be heterogenous and thus non-concordant.” Why would this a priori be reasonable? Why would malignant cells not be more clonally related?

Reply: The reviewer raises a very interesting point. Single-cell RNAseq analyses published in recent years have demonstrated that the transcriptomes of diverse normal cell types (B cells, T cells, endothelial cells etc.) are relatively homogenous and therefore can be readily used to identify these cell types independent of patient-of-origin (e.g. PMID: 27124452). In contrast, these same studies showed that malignant cells from a variety of tumors (e.g. glioblastoma, melanoma, head and neck carcinoma) are characterized by *both* intra- and inter-tumor heterogeneity in their transcriptomes, including the presence of subpopulations of malignant cells with characteristics shared between tumors, similar to what we demonstrate here. Undoubtedly, the degree of intratumor transcriptome heterogeneity will vary depending on the tumor type. Indeed, a recent paper (ref. 22) showed substantially less cell-cell concordance among TNBC cells compared to other breast cancer subtypes, which correlates with high levels of ongoing mutational diversification known to occur in TNBC. Future single-cell analyses on additional tumor types will be needed to fully characterize the range of transcriptome intratumor heterogeneity observed in human cancers. We have revised the manuscript to make this point more explicit (p. 9).

In the textual description of Fig. 4, it would be helpful to have some sort of estimate of the increased predictive power beyond just a statement of statistical significance. How much better does this do than, say, just the straight bulk expression profile?

Reply: This is an excellent point. Interestingly, few bulk signatures have been associated with survival in TNBC. Most notably, although the TNBCtype signatures (Fig. 3d) have been widely explored in this disease, they have not been convincingly shown to predict survival. Additionally, the intrinsic basal signature is thought to define an important subtype of TNBC (PMID: 22495314), and so we analyzed the intrinsic basal signature in the METABRIC cohort. As shown in Figure R4 and in new Sup. Fig. 17, unlike the cluster 2 signature (Fig. 4C), the intrinsic basal signature is not associated with patient outcome in this cohort. Finally, we conducted a similar analysis using the signatures defining each of the other cell clusters (1, 3, 4, 5) identified in this manuscript. As shown in new Sup. Fig. 16 and the response to Reviewer 1 (point 5), none of these signatures are associated with clinical outcomes in TNBC. These points have been integrated into the Results p.14 and Discussion p.17 and p.18.

Figure R4. No association of the intrinsic basal (bulk TNBC-derived) gene expression signature with survival, shown by Kaplan-Meier analysis of TNBC patients in the METABRIC cohort.

REVIEWERS' COMMENTS:

Reviewer #1 (Remarks to the Author):

The authors have sufficiently addressed all of my previous concerns. In our prior review we mentioned that single cell sequencing has been performed on breast tumors, however the authors make a good point that this is the first demonstration that patient outcomes can be associated with a subclonal population of single cells within TNBC. Overall the manuscript is much improved and will be of broad interest to the cancer field. Specific

Comments.

1. We appreciate the authors performing the analysis in Figure R1 A-D. Please include figures Figure R1A-D in the supplemental material and briefly mention the analysis and conclusions as they pertain to mature luminal and basal in the normal signatures and LAR, M and BL2 in the TNBC subtypes.

Reviewer #2 (Remarks to the Author):

The authors have nicely addressed my comments.

Re: Final revisions for manuscript NCOMMS-18-01971A

Reply to Reviewer Comments:

REVIEWERS' COMMENTS:

Reviewer #1 (Remarks to the Author):

The authors have sufficiently addressed all of my previous concerns. In our prior review we mentioned that single cell sequencing has been performed on breast tumors, however the authors make a good point that this is the first demonstration that patient outcomes can be associated with a subclonal population of single cells within TNBC. Overall the manuscript is much improved and will be of broad interest to the cancer field. Specific

Comments.

1. We appreciate the authors performing the analysis in Figure R1 A-D. Please include figures Figure R1A-D in the supplemental material and briefly mention the analysis and conclusions as they pertain to mature luminal and basal in the normal signatures and LAR, M and BL2 in the TNBC subtypes.

Reviewer #2 (Remarks to the Author):

The authors have nicely addressed my comments.

Reply:

We are pleased that the reviewers felt the paper was suitably revised and we thank the reviewers for the positive comments. We have now included the figures presented in the original response to reviewers (Figures R1A-D) as new Supplementary Figure 10. We have also mentioned these analyses and conclusions in the main text (p. 11).